# DETECTING UNKNOWN OBJECTS VIA ENERGY-BASED SEPARATION FOR OPEN WORLD OBJECT DETECTION

## ABSTRACT

In this work, we tackle the problem of Open World Object Detection (OWOD). This challenging scenario requires the detector to incrementally learn to classify given known objects without forgetting while identifying unknown objects without supervision. Previous OWOD methods have enhanced the unknown discovery process and employed memory replay to mitigate catastrophic forgetting. However, since existing methods heavily rely on the detector's known class prediction information for detecting unknown objects, they struggle to effectively learn and recognize unknown object representations. Moreover, while memory replay mitigates forgetting of old classes, it often sacrifices the knowledge of newly learned classes. To resolve these limitations, we propose DEUS (**De**tecting **U**nknowns via energy-based **S**eparation), a novel framework that addresses the challenges of Open World Object Detection. DEUS consists of ETF-Subspace Unknown Separation (EUS) and an Energy-based Known Distinction (EKD) loss. EUS leverages ETF-based geometric properties to create orthogonal subspaces, enabling cleaner separation between known and unknown object representations and leverages energies from both spaces to better capture distinct patterns of unknown objects, in contrast to prior energy-based approaches that consider only the energy within the known space. Furthermore, EKD loss enforces the separation between previous and current classifiers, thus minimizing knowledge interference between previous and newly learned classes during memory replay. We thoroughly validate DEUS on OWOD benchmarks, demonstrating outstanding performance improvements in unknown detection while maintaining competitive known class performance.

## 1 INTRODUCTION

Object Detection, a foundational task in computer vision, has achieved significant advances with the progress of deep learning (Fang et al., 2021; Girshick et al., 2014; Misra et al., 2021; Sun et al., 2021). However, traditional object detection approaches generally follow a closed-set paradigm, where the detector is restricted to recognizing only predefined classes during training. This closed-set setting hinders the detector from identifying objects that have not been encountered. To relax this restriction, Joseph *et al.* (Joseph et al., 2021) introduced a new scenario, called Open World Object Detection (OWOD), in which the detector continuously learns annotated known objects while identifying unannotated objects as unknown. In this challenging scenario, when annotations for previously unknown objects become available, the detector must be incrementally updated to recognize unknown objects as known classes. Since supervision for unknown objects is not available in OWOD, the detector faces challenges in learning knowledge for unknown objects.

To address this, prior works (Joseph et al., 2021; Ma et al., 2023b; Gupta et al., 2022; Ma et al., 2023a) propose an unknown discovery process, which utilizes the detector to assign pseudo-labels to specific regions in the background as unknowns. However, since this selection relies on the detector's current representations, it frequently selects partial areas of known objects or true background regions. This produces weak semantic pseudo-labels that blend known and unknown features, hindering effective discrimination. Since the model continuously learns from its own generated pseudo-labels during training, detecting higher-quality unknowns has become crucial in OWOD. To address this challenge, several approaches (Liang et al., 2023; Du et al., 2022; Zhang et al., 2025) integrate energy-based methods (Liu et al., 2020) by using energy scores to better identify unknown regions. However, these methods also generally consider energy only within the known space, thus rely-

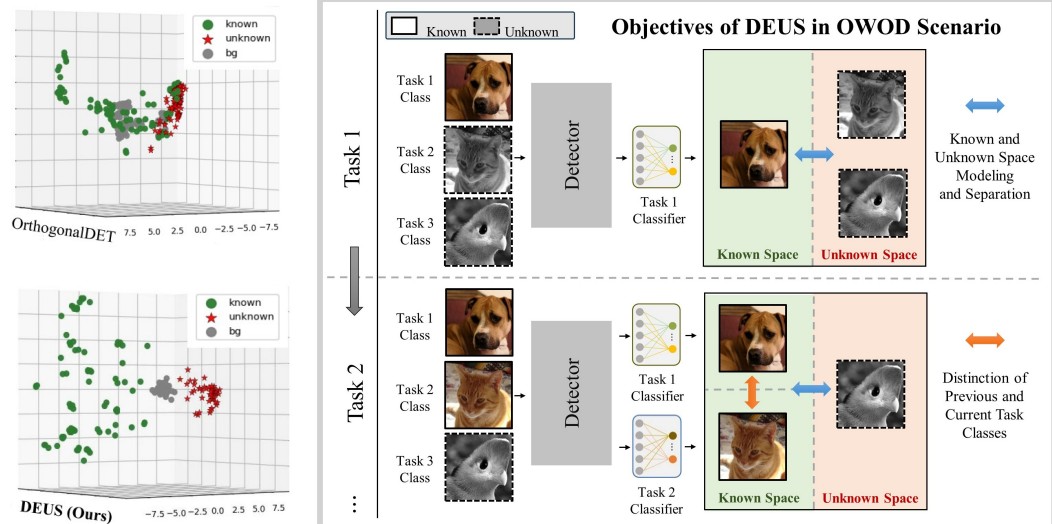

(a) Feature visualization by PCA.

(b) Objectives of the proposed method (DEUS) in the OWOD scenario.

Figure 1: Motivation and objectives of DEUS. (a) Feature visualization by PCA results comparison between baseline and DEUS. (b) Objectives of the proposed method (DEUS) in the OWOD scenario: separating known from unknown proposals while distinguishing previous and current known classes during memory replay.

ing predominantly on known representations and the detector's known classification results. Such known space-only modeling approaches push away non-known objects from the known region but lack proper constraints to prevent unknown objects from being confused with background regions or vice versa, resulting in confused representations in the feature space as shown in the top of Figure 1a. This leads to many unknown objects being overlooked or misclassified, preventing effective learning of unknown representation.

Another issue in existing methods is that there is a trade-off between the performances of previous and current classes when learning new classes. In OWOD, while the detector learns novel classes sequentially per each task, it significantly forgets the previously learned classes, *i.e.*, catastrophic forgetting arises. To address this, previous methods commonly adopt memory replay (Mermillod et al., 2013; Verwimp et al., 2021; Bonicelli et al., 2022) to retain old knowledge while learning new classes. During memory replay, the detector is fine-tuned on both old and new classes to preserve old knowledge. Even though this replay effectively alleviates the forgetting issue of old classes, existing memory replay methods lack explicit regularization to prevent cross-influence between old and new classes during training. As growing task complexity increases the number of classes to be jointly optimized, this cross-influence becomes more severe, hindering effective learning of new classes while preserving knowledge of previously learned ones.

In this paper, we propose **DEUS**, a novel OWOD framework for **De**tecting **U**nknown objects via energy-based **S**eparation. DEUS consists of **E**TF-Subspace **U**nknown **S**eparation (**EUS**) and an **E**nergy-based **K**nown **D**istinction (**EKD**) loss. As illustrated in Figure 1b, DEUS aims to address the two aforementioned challenges in the OWOD scenario by separating known and unknown proposals using EUS, and simultaneously distinguishing previous and current known classes with EKD. First, EUS creates distinct known and unknown feature spaces to more effectively identify unknown objects. Unlike existing energy-based methods (Liu et al., 2020; Liang et al., 2023; Du et al., 2022) that rely on a single known space (e.g., known classifier nodes), we jointly consider energies from two distinct Simplex-ETF subspaces—one for the known space and one for the unknown space. During training, ground-truth–matched known proposals are encouraged to attain high scores in the known subspace, while pseudo-labeled unknown proposals are encouraged to attain high scores in the unknown subspace (and relatively lower scores in the opposite subspace) and background proposals are guided toward the boundary region between the two subspaces. This bi-subspace energy learning guides features to naturally align with their respective spaces as shown in the bottom of Figure 1a and enables the detector to capture discriminative knowledge, thereby enabling effective distinction of each proposal.

Second, EKD is designed to alleviate the trade-off issue between the performances of old and new classes during memory replay. To compute energy scores separately, we partition known classifiers into old and new node classifiers. Here, higher energy scores indicate a stronger affinity to the corresponding classifier. For objects from old classes, the EKD loss encourages higher energy scores from the old classifier and lower scores from the new classifier. Conversely, for new class objects, it encourages stronger responses from the new classifier and weaker responses from the old classifier. By training the model to distinctly respond to old and new objects through these energy-based constraints, the EKD loss minimizes cross-influence between old and new classes during memory replay, enabling effective continual learning. Through comprehensive experiments, we validate the effectiveness of DEUS, which achieves significantly improved unknown recall performance while balancing the learning of old and new classes during memory replay.

Our contributions can be summarized as follows:

- We propose **De**tecting **U**nknown objects via energy-based **S**eparation (DEUS), a novel OWOD framework that addresses two challenging issues in OWOD, limited unknown representation learning and cross-influence between old and new classes.

- We introduce **ETF**-Subspace **U**nknown **S**eparation (EUS), the first approach to modeling geometrically separated distinct spaces and utilizing energy to separate known and unknown objects, helping to capture the knowledge of unknown objects and effectively discern unknowns from known or background.

- We design a new **E**nergy-based **K**nown **D**istinction (EKD) loss to alleviate the cross-influence between old and new classes during memory replay. This allows the detector to focus more on training each class set, enhancing the overall known performance.

- Experiments show that DEUS achieves state-of-the-art unknown recall across all benchmarks and tasks, while maintaining superior known mAP performance as the number of learned classes grows, demonstrating effectiveness in both unknown detection and continual learning.

## 2 RELATED WORK

### 2.1 OPEN WORLD OBJECT DETECTION

Joseph *et al.* (Joseph et al., 2021) introduced Open World Object Detection (OWOD) to address the limitations of traditional closed-set object detection. OWOD faces challenges, as detectors often confuse known and unknown representation knowledge. Prior works have attempted to enhance unknown discovery by improving pseudo-labeling and objectness. Joseph *et al.* (Joseph et al., 2021) used an RPN-based detector with an energy-based unknown identifier (EBUI), which required additional weak supervision of unknown objects. Gupta *et al.* (Gupta et al., 2022) applied attention-driven matching for pseudo-labeling, while Ma *et al.* (Ma et al., 2023b) proposed label transfer learning and annealing-based scheduling to separate known from unknown representation knowledge. Ma *et al.* (Ma et al., 2023a) decoupled localization and identification, introducing self-adaptive pseudo-labeling. Zohar *et al.* (Zohar et al., 2023) adopted a normal distribution for class-agnostic objectness, and Sun *et al.* (Sun et al., 2024) reduced the correlation between objectness and class predictions via orthogonalization. However, due to the lack of supervision for unknown objects, prior works have not focused on learning representations specific to unknowns.

### 2.2 ENERGY SCORE

Energy-based methods (Liu et al., 2020) have recently been widely adopted for out-of-distribution detection. The energy score, computed as the negative log-sum-exponential of logits, provides a unified measure for distinguishing in-distribution from out-of-distribution samples. Park *et al.* (Park et al., 2025) introduced an energy-guided discovery to identify novel categories within unlabeled data. In unknown object detection, Liang *et al.* (Liang et al., 2023) proposed a negative energy suppression loss to filter out non-object samples, while Du *et al.* (Du et al., 2022) and Zhang *et al.* (Zhang et al., 2025) introduced energy-based uncertainty regularization to model the uncertainty between known and unknown objects. In Open World Object Detection, Joseph *et al.* (Joseph et al., 2021) proposed an energy-based classifier to distinguish known from unknown objects. However, existing energy-based approaches in OWOD primarily rely on known class predictions, lacking

explicit modeling of unknown representations, which leads to confusion between unknown objects and background regions and misclassifying known object parts as unknowns.

## 3 METHOD

We propose DEUS, which effectively addresses the key challenges in Open World Object Detection. In Sec 3.1, we first introduce the problem definition of Open World Object Detection. We then describe the pipeline of the base model in Sec 3.2 for better understanding. From Sec 3.3, we propose ETF-Subspace Unknown Separation (EUS) technique that models geometrically distinct known and unknown subspaces based on Equiangular Tight Frame (ETF) (Papyan et al., 2020) and uses energy to guide objects to their respective spaces for effective separation. Finally, in Sec 3.4, we introduce an Energy-based Known Distinction (EKD) loss to balance the learning of old and new classes during memory replay.

### 3.1 PROBLEM DEFINITION

In an Open World Object Detection (OWOD) task, a total of $T$ incremental tasks are sequentially given. In the $t$-th task, where $t \in \{1, \ldots, T\}$, the detector is trained on dataset $\mathcal{D}_t = \{(\mathcal{I}_i^t, \mathcal{Y}_i^t)\}_{i=1}^N$ consisting of $N$ images, where $\mathcal{I}_i^t$ denotes the $i$-th input image and $\mathcal{Y}_i^t = \{c_j, b_j\}_{j=1}^{J_i}$ contains $J_i$ annotations for known objects. Here, $c_j$ denotes the class label, which belongs to the known class set $\mathcal{K}_t = \{1, 2, \ldots, C\}$ where $C$ denotes the number of known classes at task $t$, and $b_j = [x_j, y_j, w_j, h_j]$ denotes the bounding box. The detector trained on task $t$ can identify known objects and detect objects from the unknown class set $\mathcal{U}_t = \{C + 1, \ldots\}$ as unknowns. In the following task $t + 1$, a subset of the unknown class set, $\mathcal{U}' = \{C + 1, \ldots, C + n\}$, is labeled and merged into the updated known class set $\mathcal{K}_{t+1} = \mathcal{K}_t \cup \mathcal{U}'$, while the remaining unknown classes for task $t + 1$ are given by $\mathcal{U}_{t+1} = \mathcal{U}_t \setminus \mathcal{U}'$. By repeating this process, the detector learns new classes incrementally, expanding its knowledge of known classes while continuously identifying unseen classes as unknowns.

### 3.2 PIPELINE OF THE BASE MODEL

We adopt OrthogonalDet (Sun et al., 2024) as the base model due to its promising performance in identifying known objects in OWOD. Given an input image, the image backbone extracts a feature map and the detector obtains object proposal features $f \in \mathbb{R}^d$ from the feature map using RoI pooling (Girshick, 2015), where $d$ is the feature dimension. The extracted proposal feature $f$ is fed into the bounding box ($\mathcal{F}_{bbox}$), objectness ($\mathcal{F}_{obj}$), and classification ($\mathcal{F}_{cls}$) branches to get:

$$z_{obj} = \mathcal{F}_{obj}(||f||), \ z_{cls} = \mathcal{F}_{cls}(\frac{f}{||f||}), \ z_{bbox} = \mathcal{F}_{bbox}(f), \tag{1}$$

where $z_{obj} \in \mathbb{R}$ denotes the objectness score of the proposal $f$, and $z_{cls} \in \mathbb{R}^{C+1}$ denotes the logits for $C + 1$ classes, which include $C$ known class nodes and a single unknown node, and $z_{bbox} \in \mathbb{R}^4$ denotes the bounding box regression outputs. Using these outputs, the base model forms joint class probabilities for each proposal as $p_{jt} = \mathrm{softmax}(z_{cls}) \cdot \sigma(z_{obj})$, where $p_{jt}$ denotes joint probabilities, and $\sigma(\cdot)$ denotes the sigmoid function. During training, these joint probabilities are converted back to logits by $z_{jt} = \log \frac{p_{jt}}{1 - p_{jt}}$ for classification loss. Given the ground truth label as a one-hot vector $y_{gt}$, the classification loss is computed as follows:

$$\mathcal{L}_{\mathrm{cls}} = \mathcal{L}_{\mathrm{focal}}(y_{gt}, z_{jt}; \alpha, \gamma) \tag{2}$$

$$= -\alpha \, y_{gt} \left(1 - \sigma(z_{jt})\right)^\gamma \log \sigma(z_{jt}) \ - \ (1 - \alpha)(1 - y_{gt}) \left(\sigma(z_{jt})\right)^\gamma \log \left(1 - \sigma(z_{jt})\right), \tag{3}$$

where $\mathcal{L}_{\mathrm{focal}}(\cdot)$ denotes the sigmoid focal loss (Ross & Dollár, 2017), and $\alpha$ and $\gamma$ are focal loss hyperparameters. For the bounding box branch, the regression loss is computed using $z_{bbox}$ with:

$$\mathcal{L}_{\mathrm{bbox}} = \mathcal{L}_{\mathrm{L1}}(z_{bbox}, b_{gt}) + \mathcal{L}_{\mathrm{gIoU}}(z_{bbox}, b_{gt}), \tag{4}$$

where $b_{gt}$ denotes the ground truth bounding box coordinates, $\mathcal{L}_{\mathrm{L1}}$ refers to the L1 loss, and $\mathcal{L}_{\mathrm{gIoU}}$ denotes the generalized IoU loss (Rezatofighi et al., 2019). Given the large number of proposals generated by the detector, it is essential to select appropriate proposals for training based on ground truth. To achieve this, a dynamic matcher (Wang et al., 2023) is adopted to align object proposals

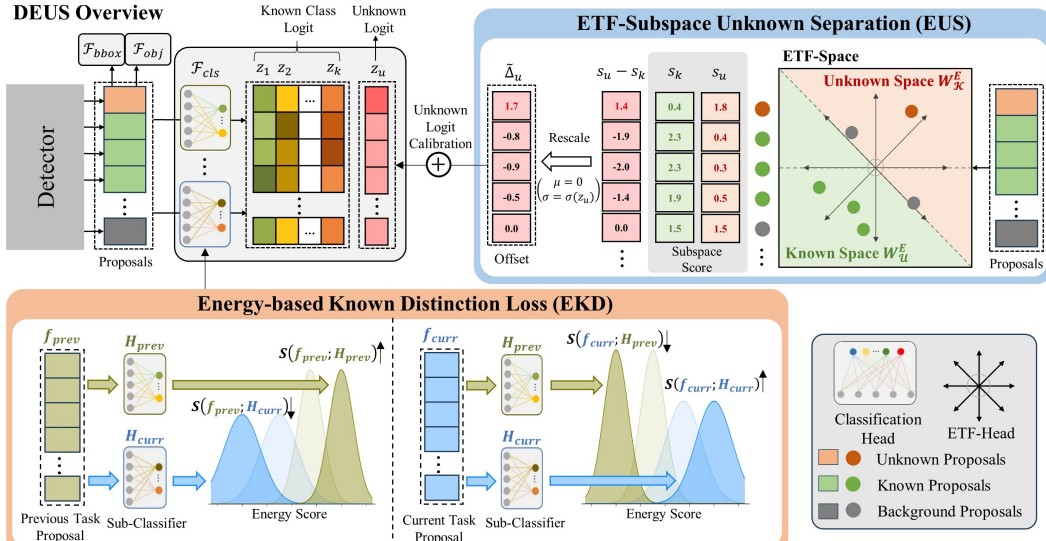

Figure 2: Overview of DEUS. ETF-Subspace Unknown Separation (EUS) utilizes a simplex ETF to construct known and unknown spaces, where each space serves as an energy module to compute space energy scores, and Energy-based Known Distinction (EKD) loss is applied during the memory replay phase, where classification branch is split into two sub-classifiers to calculate energy scores for previous and current tasks.

with ground truth, while a subset of the unmatched proposals is used as training targets for unknown objects. Furthermore, in OWOD, the sequential training of new classes leads to severe forgetting after each incremental training phase. To address this issue, memory replay is employed to mitigate catastrophic forgetting. For memory replay, samples are stored in a memory buffer and utilized in the subsequent replay phase.

### 3.3 ETF-Subspace Unknown Separation

Existing OWOD methods have typically defined unknown objects based solely on the detector's known class predictions, without considering distinctive representations specific to unknown objects. This approach leads to missing unknown objects or misclassifying them as background regions. To address this, we propose the ETF-Subspace Unknown Separation (EUS), which explicitly builds two ETF-aligned subspaces (known / unknown) and guides each proposal toward its respective subspace. To this end, we first construct geometrically separated known and unknown spaces using a Simplex ETF. The simplex ETF is defined mathematically as follows:

$$W^E = \sqrt{\frac{K}{K-1}} \left( I_K - \frac{1}{K} \mathbf{1}_K \mathbf{1}_K^\top \right) Q, \tag{5}$$

where $W^E \in \mathbb{R}^{K \times d}$ represents the ETF basis matrix containing $K$ equiangular vectors, $\mathbf{1}_K$ is the all-ones vector, $Q \in \mathbb{R}^{K \times d}$ is an orthonormal matrix (i.e., $Q^\top Q = I_d$), $K$ is the number of ETF basis vectors. From the definition of simplex ETF, we define known and unknown spaces as follows:

$$W_{\mathcal{K}}^E = [W_1^E; \ldots; W_{\frac{K}{2}}^E] \in \mathbb{R}^{\frac{K}{2} \times d}, \qquad W_{\mathcal{U}}^E = [W_{\frac{K}{2}+1}^E; \ldots; W_K^E] \in \mathbb{R}^{\frac{K}{2} \times d}, \tag{6}$$

where $W_{\mathcal{K}}^E$ and $W_{\mathcal{U}}^E$ denote non-overlapping subspaces from the simplex ETF, and these bases are fixed and non-learnable.

Using the separated subspaces, the detector computes the energy with respect to the subspace. Each space functions as an energy module to compute the *Helmholtz free energy* (Liu et al., 2020) for a proposal feature $f$:

$$E^{\mathcal{K}}(f) = -\log \sum_{i=1}^{K/2} \exp\left(W_{\mathcal{K},i}^E \cdot f\right), \qquad E^{\mathcal{U}}(f) = -\log \sum_{i=1}^{K/2} \exp\left(W_{\mathcal{U},i}^E \cdot f\right). \tag{7}$$

where $E^{\mathcal{K}}(f)$ and $E^{\mathcal{U}}(f)$ denote the known and unknown energies, respectively, while $W_{\mathcal{K},i}^E \cdot f$ and $W_{\mathcal{U},i}^E \cdot f$ represent the projections of feature $f$ onto the $i$-th ETF basis vectors in the known and

unknown subspaces. Then, we define subspace scores by negating energies:

$$s_k(f) = -E^{\mathcal{K}}(f), \qquad s_u(f) = -E^{\mathcal{U}}(f), \tag{8}$$

where $s_k$ and $s_u$ denote the known and unknown scores indicating whether $f$ belongs to the known and unknown subspaces, respectively, where higher value implies a stronger association with the corresponding subspace. We then define the unknown offset as:

$$\Delta_u(f) = s_u(f) - s_k(f), \tag{9}$$

where $\Delta_u(f)$ represents how much higher the unknown subspace score is compared to the known subspace score for feature $f$. To guide known and unknown proposals toward their respective subspaces, EUS employs a loss function consisting of two complementary terms. First, we use an energy-based margin loss on the unknown offset $\Delta_u$:

$$\mathcal{L}_{\text{energy}} = \mathbb{E}_f \left[ \begin{cases} \max\left(0, m + \Delta_u(f)\right)^2 & \text{if } f \text{ is GT-matched} \\ \max\left(0, m - \Delta_u(f)\right)^2 & \text{if } f \text{ is pseudo-unknown} \\ 0 & \text{otherwise} \end{cases} \right], \tag{10}$$

where $m$ is a hyperparameter for the minimum margin gap. This loss enforces a margin between known and unknown scores by driving $\Delta_u(f) \leq -m$ for known and $\Delta_u(f) \geq m$ for pseudo-unknown proposals. For more stable convergence during training, we additionally adopt a focal loss (equation 3) on the subspace scores $[s_k, s_u]$:

$$\mathcal{L}_{\text{subspace}} = \mathcal{L}_{\text{focal}}(t, [s_k, s_u]; \alpha, \gamma), \tag{11}$$

where target one-hot vector is $t = [1, 0]$ for GT-matched queries, $t = [0, 1]$ for pseudo-unknown queries, and $t = [0, 0]$ for background. With this objective, known proposals are guided toward the known ETF subspace, while unknown proposals are guided toward the complementary unknown ETF subspace. Background queries receive negative targets for both energy scores, encouraging them toward the boundary region between the two subspaces. The final EUS objective is the sum of the two terms:

$$\mathcal{L}_{\text{EUS}} = \mathcal{L}_{\text{energy}} + \mathcal{L}_{\text{subspace}}. \tag{12}$$

While energy loss provides the primary mechanism for known-unknown separation, the subspace loss guides known, unknown, and background proposals to their respective regions stably.

At inference time, following energy-based unknown scoring (Liu et al., 2020), we first calculate the unknown logit $z_u = \log \sum_{c \in C} \exp(z_c)$, where $z_c$ represents the logit for known class $c$ from the classification head $z_{cls}$. Next, to reflect the subspace scores, we calibrate $z_u$ with the unknown offset $\Delta_u(f)$. We form a calibration term that decreases the logit for known proposals and increases it for unknown proposals by standardizing $\Delta_u(f)$ per image across proposals, $\tilde{\Delta}_u(f) = (\Delta_u(f) - \mu_{\Delta_u})/\sigma_{\Delta_u}$, and then rescaling it by the standard deviation of $z_u$ (denoted as $\sigma_{z_u}$) to match the scale. Accordingly, the final unknown logit is given by

$$z_u' = z_u + \sigma_{z_u} \tilde{\Delta}_u(f). \tag{13}$$

This calibration enhances the logit for unknown proposals while suppressing it for known proposals. For background proposals, which comprise most of the proposals, near-zero calibration values are obtained. In contrast to previous energy-based approaches that depend solely on the detector's known class head prediction, EUS explicitly projects features onto two subspaces to effectively capture not only known features but also unknown representations, which prior works overlook.

### 3.4 ENERGY-BASED KNOWN DISTINCTION LOSS

In OWOD, memory replay is commonly used to mitigate catastrophic forgetting. Although this technique helps the detector to retain the knowledge of old classes, as the number of known classes grows and the classification problem becomes more complex, the cross-influence between old and new classes makes it difficult to preserve old knowledge and learn new concepts simultaneously. To address this, we propose an *Energy-based Known Distinction* (EKD) loss that reduces cross-influence during memory replay by explicitly separating the old and new classifiers and guiding each proposal to its corresponding classifier using energy scores. Concretely, we split the known class

classifier into two sub-classifiers: the previous task sub-classifier $H_{\text{prev}}$ and the current task sub-classifier $H_{\text{curr}}$, each handling its own classes. For each proposal feature $f$, we define the negative energy-based score as follows:

$$S(f; H) \;=\; -E(f; H) \;=\; \log \sum_{c=1}^{C_H} \exp\big(z_c(f; H)\big), \tag{14}$$

where $H \in \{H_{\text{prev}}, H_{\text{curr}}\}$ denotes a sub-classifier, $z_c(f; H)$ is the logit for class $c$, and $C_H$ is the number of classes handled by sub-classifier $H$. Similar to the energy scores defined in Sec. 3.3, a larger $S$ indicates lower energy (i.e., stronger affinity) with the sub-classifier $H$, as higher logits in the log-sum-exponential formulation naturally correspond to stronger confidence within that classifier's domain. To minimize cross-interference, we encourage each sub-classifier to respond more strongly to its corresponding task samples. Specifically, let $f_{\text{prev}}$ and $f_{\text{curr}}$ denote proposals from previous and current tasks, respectively. We expect proposals from previous tasks to have higher energy scores with $H_{\text{prev}}$ than with $H_{\text{curr}}$, that is, $S(f_{\text{prev}}; H_{\text{prev}}) > S(f_{\text{prev}}; H_{\text{curr}})$, and vice versa for current task proposals. To this end, we design the EKD loss that enforces these preferences via pairwise loss as follows:

$$\mathcal{L}_{\text{prev}} = \log\Big(1 + \exp\big[S(f_{\text{prev}}; H_{\text{curr}}) - S(f_{\text{prev}}; H_{\text{prev}})\big]\Big), \tag{15}$$

$$\mathcal{L}_{\text{curr}} = \log\Big(1 + \exp\big[S(f_{\text{curr}}; H_{\text{prev}}) - S(f_{\text{curr}}; H_{\text{curr}})\big]\Big), \tag{16}$$

$$\mathcal{L}_{\text{EKD}} = \mathcal{L}_{\text{prev}} + \mathcal{L}_{\text{curr}}. \tag{17}$$

This contrastive loss reduces the interference of $H_{\text{curr}}$ in previous task proposals and of $H_{\text{prev}}$ in current task proposals, effectively mitigating the cross-influence during memory replay. The overall training objective combines all components:

$$\mathcal{L}_{\text{total}} = \mathcal{L}_{\text{cls}} + \mathcal{L}_{\text{bbox}} + \mathcal{L}_{\text{EUS}} + \mathcal{L}_{\text{EKD}}, \tag{18}$$

where $\mathcal{L}_{\text{EKD}}$ is applied only during memory replay phases when training on incremental tasks. To sum up, EUS first constructs the simplex ETF aligned known and unknown subspaces, and trains the detector to align features with their respective subspaces while leveraging energy scores of both known and unknown subspaces. As a result, the detector achieves stronger unknown detection and effective discrimination between known and unknown objects. Moreover, the EKD loss aims to strike a balance between old and new classes during memory replay. EKD guides the detector to reduce cross-influence by contrastively learning energy scores from each sub-classifier.

## 4 EXPERIMENTS

### 4.1 EXPERIMENTAL SETTINGS

**Datasets and Metrics.** We evaluated our method, DEUS, on M-OWODB (Joseph et al., 2021) and S-OWODB (Gupta et al., 2022). M-OWODB refers to the "Superclass-Mixed OWOD Benchmark" and consists of the COCO (Lin et al., 2014) and PASCAL VOC (Everingham et al., 2010) datasets, grouped into four sets of non-overlapping tasks. S-OWODB represents the "Superclass-Separated Benchmark" and uses only the COCO dataset, also divided into four non-overlapping tasks. We validate DEUS by considering both known and unknown classes. For known classes, we used mean average precision (mAP) as the metric, measuring mAP for previously learned classes, currently learned classes, and all known classes. For unknown classes, we used the recall (U-Rec) as the main metric to evaluate the performance of detecting unknown objects. To measure the overall performance, considering both known and unknown objects, we adopted the harmonic mean (H-score), combining the mAP of known and the recall of unknown classes.

**Implementation Details.** We used OrthogonalDet (Sun et al., 2024) as our base model. OrthogonalDet is based on a Fast R-CNN (Girshick, 2015) and a ResNet-50 (He et al., 2016) pre-trained on ImageNet (Russakovsky et al., 2015). The weight of the ETF and EKD loss is set to 1.0. We set the number of $K$ for the Simplex ETF spaces to 128, constructing both known and unknown spaces with 64 vectors each. Our implementation is based on MM-Detection (Chen et al., 2019). For further implementation details, please refer to the supplementary materials (see Appendix A).

Table 1: Experimental results on M-OWODB (top) and S-OWODB (bottom). Results are reported in terms of mean average precision (mAP) for known classes, unknown class recall (U-Rec), and harmonic score (H-Score). The best performance is highlighted in bold, with the second-best performance underlined. † denotes reproduced results after correcting M-OWODB annotation duplication bug identified in (Yavuz & Güney, 2024), which may differ from the originally reported numbers.

| Task IDs | Task 1 | | | Task 2 | | | | | Task 3 | | | | | Task 4 | | |
|---|---|---|---|---|---|---|---|---|---|---|---|---|---|---|---|---|
| Method | Current mAP | U-Rec | H-Score | Previous mAP | Current mAP | Known mAP | U-Rec | H-Score | Previous mAP | Current mAP | Known mAP | U-Rec | H-Score | Previous mAP | Current mAP | Known mAP |
| ORE (Joseph et al., 2021) | 56.0 | 4.9 | 9.0 | 52.7 | 26.0 | 39.4 | 2.9 | 5.4 | 38.2 | 12.7 | 29.7 | 3.9 | 6.9 | 29.6 | 12.4 | 25.3 |
| OW-DETR (Gupta et al., 2022) | 59.2 | 7.5 | 13.3 | 53.6 | 33.5 | 42.9 | 6.2 | 10.8 | 38.3 | 15.8 | 30.8 | 5.7 | 9.6 | 31.4 | 17.1 | 27.8 |
| CAT (Ma et al., 2023a) | 60.0 | 23.7 | 34.0 | 55.5 | 32.7 | 44.1 | 19.1 | 26.7 | 42.8 | 18.7 | 34.8 | 24.4 | 28.7 | 34.4 | 16.6 | 29.9 |
| PROB† (Zohar et al., 2023) | **66.4** | 28.3 | 39.7 | **62.6** | 39.2 | 50.9 | 26.4 | 34.8 | 49.6 | 33.5 | 44.2 | 29.3 | 35.2 | 44.0 | 26.5 | 39.7 |
| OrthogonalDet† (Sun et al., 2024) | 63.4 | 24.1 | 34.9 | 58.2 | 44.0 | 51.1 | 24.7 | 33.3 | 50.9 | 40.1 | 47.3 | 28.7 | 35.7 | 49.1 | 31.5 | 44.7 |
| O1O (Yavuz & Güney, 2024) | 65.1 | 49.3 | 56.1 | 61.0 | 45.0 | 53.0 | 50.3 | 51.6 | 50.0 | 36.5 | 45.5 | 49.5 | 47.4 | 46.2 | 31.0 | 42.4 |
| OWOBJ (Zhang et al., 2025) | 61.4 | 23.6 | 34.1 | 58.4 | 34.4 | 45.7 | 23.8 | 31.3 | 44.8 | 27.8 | 38.8 | 25.1 | 30.5 | 36.4 | 20.7 | 32.0 |
| **DEUS (Ours)** | 66.2 | 65.1 | 65.6 | 61.0 | 45.7 | 53.3 | 66.2 | 59.0 | 53.4 | 43.3 | 50.1 | 69.0 | 58.0 | 50.5 | 32.8 | 46.0 |
| ORE (Joseph et al., 2021) | 61.4 | 1.5 | 2.9 | 56.5 | 26.1 | 40.6 | 3.9 | 7.1 | 38.7 | 23.7 | 33.7 | 3.6 | 6.5 | 33.6 | 26.3 | 31.8 |
| OW-DETR (Gupta et al., 2022) | 71.5 | 5.7 | 10.6 | 62.8 | 27.5 | 43.8 | 6.2 | 10.9 | 45.2 | 24.9 | 38.5 | 6.9 | 11.7 | 38.2 | 28.1 | 33.1 |
| CAT (Ma et al., 2023a) | 74.2 | 24.0 | 36.3 | 67.6 | 35.5 | 50.7 | 23.0 | 31.6 | 51.2 | 32.6 | 45.0 | 24.6 | 31.8 | 45.4 | 35.1 | 42.8 |
| PROB (Zohar et al., 2023) | 73.4 | 17.6 | 28.4 | 66.3 | 36.0 | 50.4 | 22.3 | 30.9 | 47.8 | 30.4 | 42.0 | 24.8 | 31.2 | 42.6 | 31.7 | 39.9 |
| OrthogonalDet (Sun et al., 2024) | 71.6 | 24.6 | 36.6 | 64.0 | 39.9 | 51.3 | 27.9 | 36.1 | 52.1 | 42.2 | 48.8 | 31.9 | 38.6 | 48.7 | 38.8 | 46.2 |
| O1O (Yavuz & Güney, 2024) | 72.6 | 49.8 | 59.1 | 65.3 | 44.9 | 54.6 | 51.1 | 52.8 | 49.5 | 41.5 | 46.8 | 48.1 | 47.4 | 47.3 | 42.0 | 45.9 |
| OWOBJ (Zhang et al., 2025) | 76.2 | 22.3 | 34.5 | 69.8 | 41.0 | 54.8 | 28.7 | 37.7 | 50.6 | 35.7 | 46.8 | 30.9 | 37.2 | 46.7 | 36.9 | 43.2 |
| **DEUS (Ours)** | 71.6 | 68.7 | 70.1 | 63.5 | 43.0 | 52.7 | 62.9 | 57.4 | 53.6 | 45.4 | 50.9 | 60.7 | 55.4 | 50.7 | 42.8 | 48.8 |

Table 2: Ablation study of DEUS on M-OWODB. EUS and EKD represent ETF-Subspace Unknown Separation and Energy-based Known Distinction loss, respectively. EUS aims to improve the detection performance for unknown objects, while EKD is designed to enhance the performance for known objects. The best performance is highlighted in bold, with the second-best performance underlined.

| Task IDs | | Task 1 | | | Task 2 | | | | | Task 3 | | | | | Task 4 | | |
|---|---|---|---|---|---|---|---|---|---|---|---|---|---|---|---|---|---|
| EUS | EKD | Current mAP | U-Rec | H-Score | Previous mAP | Current mAP | Known mAP | U-Rec | H-Score | Previous mAP | Current mAP | Known mAP | U-Rec | H-Score | Previous mAP | Current mAP | Known mAP |
| | | 66.0 | 36.8 | 47.2 | 58.8 | 45.3 | 52.0 | 29.0 | 37.3 | 52.7 | 43.5 | 49.7 | 30.3 | 37.6 | 49.2 | 31.3 | 44.7 |
| | ✓ | 66.0 | 36.8 | 47.2 | 59.2 | 45.9 | 52.6 | 40.0 | 45.4 | 53.6 | 43.6 | 50.3 | 38.9 | 43.9 | 50.3 | 32.7 | 45.9 |
| ✓ | | 66.2 | 65.1 | 65.6 | 58.8 | 45.0 | 51.9 | 63.8 | 57.2 | 52.6 | 42.4 | 49.2 | 69.0 | 57.5 | 48.1 | 29.7 | 43.5 |
| ✓ | ✓ | 66.2 | 65.1 | 65.6 | 61.0 | 45.7 | 53.3 | 66.2 | 59.0 | 53.4 | 43.3 | 50.1 | 69.0 | 58.0 | 50.5 | 32.8 | 46.0 |

## 4.2 EXPERIMENTAL RESULTS

We present the comparison results for M-OWODB (top) and S-OWODB (bottom) in Table 1, comparing our DEUS with previous OWOD methods (Joseph et al., 2021; Gupta et al., 2022; Ma et al., 2023a; Zohar et al., 2023; Sun et al., 2024; Zhang et al., 2025; Yavuz & Güney, 2024). In OWOD benchmarks, a fundamental trade-off exists between known mAP and unknown recall (U-Rec). Methods focusing on unknown detection often sacrifice known class performance by misclassifying known objects as unknown, while maintaining high known mAP leads to poor unknown recall. The H-Score, measuring the harmonic mean of both metrics, reflects the model's ability to accurately separate known and unknown objects. As shown in Table 1, our DEUS achieves the best H-Score performance across all tasks. In particular, DEUS shows strong unknown detection capability, achieving U-Recall scores of 65.1, 66.2, and 69.0 for Tasks 1–3, which clearly outperform other methods. These large improvements in unknown detection are achieved while maintaining competitive known mAP performance, resulting in significantly improved H-Scores that demonstrating our EUS method's superior ability to distinguish known from unknown objects. Additionally, while other methods showed degraded known mAP as tasks progress, DEUS maintains stable performance throughout the incremental learning process, showing better resistance to catastrophic forgetting through our EKD loss. From Tasks 3-4 onward, DEUS achieves the best performance in all known mAP metrics, showing that our approach successfully reduces cross-influence between previously learned and newly learned classes. To evaluate the generalizability of DEUS, we constructed a new OWOD benchmark using remote sensing images and compared it with our baseline. Detailed experimental results can be found in supplementary materials (see Appendix C).

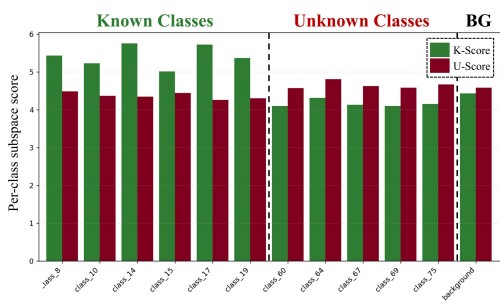

(a) Comparison of subspace scores per classes.

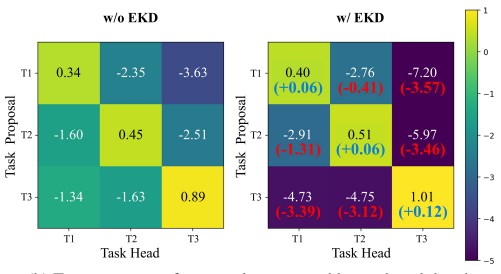

(b) Energy scores of proposals computed by each task head.

Figure 3: Analysis of DEUS. (a) Per-class subspace score comparison between known (green) and unknown (red) for queries matched with actual ground-truth objects. Note that known classes are 1-20. (b) Energy scores of proposals from each task computed by each task head (w/o vs. w/ EKD). The numbers in parentheses represent the change in values when EKD is applied.

## 4.3 ABLATION STUDY

We conducted an ablation study on the M-OWODB benchmark to validate the performance improvements contributed by each component of DEUS. Table 2 summarizes the performance comparison of each ablated case of the proposed DEUS. EUS significantly enhances U-Recall performance, as it aims to disentangle representation knowledge between known and unknown objects, effectively helping the detector capture knowledge of unknown objects. However, since EUS focuses on detecting unknown objects, the known mAP shows a slight decrease as a result of the increased number of unknown detections. In contrast, EKD consistently leads to performance improvements across all tasks regardless of whether EUS is applied. When both components are applied, DEUS achieved the best performance, with the highest H-Score on every task.

## 4.4 ANALYSIS

To show how well EUS distinguishes between known and unknown objects, we analyzed the per-class subspace scores for queries that match actual ground-truth objects using a model trained on Task 1 classes (known classes 1–20), as shown in Fig. 3a. The results show that queries matched to learned classes (1–20) consistently have higher Known scores than Unknown scores, with strong affinity toward the known subspace. In contrast, queries matched to classes not yet encountered during training (classes 21-80) exhibit higher Unknown scores, indicating proper alignment with the unknown subspace. Background proposals (those unmatched to any GT and with low objectness scores) are naturally placed in marginal scores between the two, which aligns with our expectations. This clear separation between known and unknown representations naturally aligns proposals in the feature space as visualized in bottom of Fig. 1a, enabling effective distinction between known, unknown, and background objects. Furthermore, to verify whether EKD actually increases scores for proposals belonging to each task's sub-classifier while suppressing scores for non-belonging proposals, we visualize energy scores (Eq. 14) heatmap for each task head on proposals from each task as shown in Fig. 3b. Ideally, diagonal elements (matching task-head pairs) should be high while off-diagonal elements should be low. With EKD, diagonal scores increase while off-diagonal scores are significantly suppressed compared to without EKD (e.g., T3 proposals with T1 head: -1.34 → -4.73), demonstrating that EKD effectively guides proposals towards their appropriate task heads.

## 5 CONCLUSION

In this paper, we propose DEUS, a novel framework for Open World Object Detection (OWOD). OWOD requires models to learn known classes incrementally while detecting unknown objects, which raises challenges such as limited representation learning of unknown objects and cross-influence between old and new classes during memory replay. DEUS addresses these issues through two modules: ETF-Subspace Unknown Separation (EUS), which captures representations of unknown objects by separating them from known, and Energy-based Known Distinction (EKD), which mitigates cross-influence by focusing on each class set. Nonetheless, semantic overlap between known and unknown objects remains challenging, motivating future work on more refined representation learning for unknown objects.

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
