# Supplementary Material

In this supplementary material, we provide additional details and analyses that could not be included in the main paper due to page constraints. Section A covers implementation details and benchmark settings. Section B presents further analysis of utilizing simplex ETF. Section C presents experimental results in the remote sensing domain to demonstrate generalizability of DEUS. Section D presents qualitative results.

## A  DETAILS OF EXPERIMENT SETTINGS

**Implementation Details**   We follow the implementation details of OrthogonalDet (Sun et al., 2024). OrthogonalDet is based on a Fast R-CNN (Girshick, 2015) architecture and a ResNet-50 (He et al., 2016) backbone pre-trained on ImageNet (Russakovsky et al., 2015). We used a linear classifier for the classification and Batch Normalization (Ioffe & Szegedy, 2015) for the objectness branch, following OrthogonalDet. Our DEUS framework is trained using the AdamW optimizer (Loshchilov & Hutter, 2019). RoI pooling is applied to 500 random proposals, which are then fed into the detection heads for localization, objectness, and classification training. During training, DEUS is supervised using ground-truth annotations and pseudo-labels selected based on our pseudo-labeling process in A. During inference, we omit prediction randomness by using 1,000 pre-defined object proposals, which are pruned via non-maximum suppression at an IoU threshold of 0.6. The final detections are selected using a score threshold of 0.10. For loss hyperparameters and balancing, we set the focal loss parameters $\alpha = 0.25$ and $\gamma = 2.0$. The classification loss weight is 2.0, L1 regression loss weight is 5.0, and GIoU loss weight is 2.0. Our proposed EUS and EKD loss weights are both set to 1.0, and ETF energy margin $m$ is 0.5. We used three NVIDIA RTX 4090 GPUs with a batch size of 12 per GPU, employing AMP with bfloat16 precision for efficient training and our implementation is based on MM-Detection (Chen et al., 2019) framework.

**Pseudo-labeling process.**   Following the baseline approach (Sun et al., 2024), we provide pseudo-labels for unknown objects as supervision signals to the detector. While the baseline method simply selects a fixed number $\tau$ (e.g., 20) of queries with the highest objectness scores among those unmatched with ground truth, we introduce a more sophisticated pseudo-labeling process. First, The number of pseudo-labels $k_{\mathrm{pseudo}}$ is determined proportionally to the known ground truth count, following a dynamic scaling mechanism $k_{\mathrm{pseudo}} = k_{\mathrm{gt}} \cdot \max\left(1, \frac{2\tau}{N_{\mathrm{known}}}\right)$, where $N_{known}$ represents the number of learned classes including the current task and we set $\tau = 20$. This provides more unknown supervision in early stages when fewer classes have been learned, gradually reducing as the model learns more classes. Second, we only consider objects whose bounding-box length is at least 0.5 times the minimum image size as unknown labels, filtering out excessively small or noisy detections. Lastly, we use final unknown logits (Eq.equation 13) as our criterion, selecting only objects with scores above zero as pseudo-labels to maximize avoidance of known objects. This creates a beneficial self-improving cycle: our EUS method enables better unknown detection, which provides higher-quality pseudo-labels that further enhance unknown representation learning, leading to progressively improved performance without additional unknown supervision.

**M-OWODB**   The superclass-mixed benchmark (Joseph et al., 2021) groups all Pascal VOC classes and data into the initial task, Task 1. The remaining 60 classes from MS-COCO are then divided into three incremental tasks, introducing semantic drifts. Due to the overlap between Pascal VOC and MS-COCO classes, super-categories are mixed across tasks. Consequently, while M-OWODB ensures non-overlapping classes between sequential tasks, super-categories such as Vehicles and Animals may still overlap across tasks. The left section of S.Table 1 shows the benchmark configuration of M-OWODB, where Task 1 consists of the VOC classes that can share the super-categories with subsequent tasks.

**S-OWODB**   The superclass-separated benchmark (Gupta et al., 2022) provides a stricter MS-COCO split compared to M-OWODB. While M-OWODB allows data leakage across tasks due to the inclusion of different classes from the same super-categories (*e.g.*, most classes from vehicle

S.Table 1: Benchmark Configuration for M-OWODB, S-OWODB, and RS-OWODB.

| Metrics | M-OWODB | | | | S-OWODB | | | | RS-OWODB | | | |
|---|---|---|---|---|---|---|---|---|---|---|---|---|
| | Task 1 | Task 2 | Task 3 | Task 4 | Task 1 | Task 2 | Task 3 | Task 4 | Task 1 | Task 2 | Task 3 | Task 4 |
| Classes | VOC Classes | Outdoor, Accessories, Appliances, Truck | Sports, Food | Electronic, Indoor, Kitchen, Furniture | Animals, Person, Vehicles | Outdoor, Accessories, Appliances, Furniture | Sports, Food | Electronic, Indoor, Kitchen | Baseballfield, Dam, Groundtrackfield, Stadium, Vehicle | Basketballcourt, Expressway Area, Harbor, Storagetank, Windmill | Aeroplane, Bridge, Expressway Station, Overpass, Tenniscourt | Airport, Chimney, Golffield, Ship, Trainstation |
| # of Classes | 20 | 20 | 20 | 20 | 19 | 21 | 20 | 20 | 5 | 5 | 5 | 5 |
| # of training images | 16,551 | 45,520 | 39,402 | 40,260 | 89,490 | 55,870 | 39,402 | 38,903 | 5,394 | 3,445 | 4,111 | 3,247 |
| # of training objects | 47,223 | 113,741 | 114,452 | 138,996 | 421,243 | 163,512 | 114,452 | 160,794 | 18,378 | 9,928 | 10,093 | 29,674 |
| # of test images | 10,246 | | | | 8,877 | | | | 11,738 | | | |
| # of test objects | 14,976 | 4,966 | 4,826 | 6,039 | 17,786 | 7,159 | 4,826 | 7,010 | 8,212 | 32,695 | 20,614 | 37,967 |

and animal super-categories are introduced in Task 1, while related classes such as truck, elephant, bear, zebra, and giraffe appear in Task 2), S-OWODB groups all categories within a super-category into a single task rather than spreading them across tasks. As shown in middle of S.Table 1, Task 1 contains all related classes from Animals, Person, and Vehicles, while Task 2 includes Appliances, Accessories, Outdoor, and Furniture. This strict separation by super-categories makes S-OWODB a more challenging OWOD benchmark.

**RS-OWODB** To further evaluate the generalizability of DEUS, we introduce a new benchmark setting called RS-OWODB (Remote Sensing OWODB) as shown in the right side of S.Table 1. Unlike M-OWODB and S-OWODB, which focus on natural images, RS-OWODB utilizes remote sensing images. This benchmark is constructed using the DIOR (Li et al., 2019) datasets with each task consisting of 5 classes. The benchmark maintains balanced data distribution across tasks, with the number of images and object instances evenly distributed. This benchmark provides an additional evaluation environment for the OWOD scenario in the remote sensing domain.

# B   ANALYSIS ON SIMPLEX ETF

Neural collapse (Papyan et al., 2020) is a phenomenon in which the activations of the last layer and the classifier vectors form a simplex equiangular tight frame (ETF). The simplex ETF consists of $K$ vectors in $\mathbb{R}^d$, where all vectors have equal $\ell_2$ norm, and any pair has an inner product of $-\frac{1}{K-1}$. The fixed simplex ETF is an ideal classifier structure due to its equiangular and consistent magnitude. Motivated by these properties, we adopt the fixed simplex ETF to construct both known and unknown spaces. The equiangular and equal magnitude properties of the $K$ vectors in the simplex ETF allow subsets of these vectors to form non-overlapping spaces, each with a margin of $-\frac{1}{K-1}$. Using the simplex ETF, EUS effectively guides proposal into the appropriate known and unknown spaces.

To validate the effectiveness of the known and unknown spaces constructed using the simplex ETF, we conducted an ablation study to evaluate the impact of the number of $K$ vectors used in the simplex ETF. Increasing the number of $K$ helps create known and unknown spaces with sufficient vectors to represent each space and improve the model's ability to capture unknown patterns, resulting in generally increased U-Rec performance. However, this enhanced unknown detection capability leads a slight decrease in Known mAP due to the increased number of unknown detections, though the difference is not significant. As shown in S.Table 2, we empirically identified the optimal value for $K$. We observed that larger $K$ values consistently improve unknown recall performance, with $K = 128$ achieving the highest U-Rec of 69.0. Although there is a minor trade-off in Known mAP, the overall H-Score, which represents the harmonic mean between Known mAP and U-Rec, reaches its peak at $K = 128$. Therefore, we select $K = 128$ as our final configuration to achieve the best balance between known and unknown object detection performance.

S.Table 2: Analysis of the number of $K$ for simplex ETF on M-OWODB. We conducted an ablation study to determine the optimal number of $K$ for the simplex ETF. The best performance is highlighted in bold, and the second best is underlined.

| Task IDs | Task 1 | | | Task 2 | | | | | Task 3 | | | | |
|---|---|---|---|---|---|---|---|---|---|---|---|---|---|
| # of $K$ | Current mAP | U-Rec | H-Score | Previous mAP | Current mAP | Known mAP | U-Rec | H-Score | Previous mAP | Current mAP | Known mAP | U-Rec | H-Score |
| w/o EUS | 66.0 | 36.8 | 47.2 | 59.2 | 45.9 | 52.6 | 40.0 | 45.4 | 53.6 | 43.6 | 50.3 | 38.9 | 43.9 |
| $K$=32 | 66.4 | 62.9 | 64.6 | 61.0 | 46.3 | 53.6 | 63.8 | 58.3 | 53.8 | 43.6 | 50.4 | 65.5 | 57.0 |
| $K$=64 | 66.0 | 64.4 | 65.2 | 60.6 | 45.5 | 53.0 | 64.0 | 58.0 | 53.4 | 43.1 | 50.0 | 66.0 | 56.9 |
| **K=128** | 66.2 | 65.1 | 65.6 | 61.0 | 45.7 | 53.3 | 66.2 | 59.0 | 53.4 | 43.3 | 50.1 | 69.0 | 58.0 |

S.Table 3: Experimental results on RS-OWODB. Results are reported in terms of mean average precision (mAP) for known classes, unknown class recall (U-Rec), and harmonic score (H-Score). The best performance is highlighted in bold.

| Task IDs | Task 1 | | | Task 2 | | | | | Task 3 | | | | | Task 4 | | |
|---|---|---|---|---|---|---|---|---|---|---|---|---|---|---|---|---|
| Method | Current mAP | U-Rec | H-Score | Previous mAP | Current mAP | Known mAP | U-Rec | H-Score | Previous mAP | Current mAP | Known mAP | U-Rec | H-Score | Previous mAP | Current mAP | Known mAP |
| OrthogonalDet (Sun et al., 2024) | 65.4 | 23.7 | 34.8 | 55.2 | 69.4 | 62.3 | 8.9 | 15.6 | 63.9 | 59.5 | 62.4 | 9.3 | 16.2 | 62.4 | 69.5 | 64.2 |
| **DEUS (Ours)** | 67.7 | 58.1 | 62.5 | 58.1 | 73.1 | 65.6 | 28.1 | 39.4 | 66.9 | 64.3 | 66.0 | 29.7 | 40.9 | 66.4 | 74.0 | 68.3 |

# C  EXPERIMENTAL RESULTS ON RS-OWODB

To evaluate the generalizability of DEUS beyond natural images, we conducted experiments on RS-OWODB using remote sensing images from the DIOR dataset. This different domain provided a more challenging evaluation environment with different visual characteristics, object scales, and spatial distributions. As shown in S.Table 3, DEUS demonstrated superior performance compared to our base model (OrthogonalDet), across all tasks in RS-OWODB. In Task 1, DEUS achieved 67.7 current mAP and 58.1 U-Rec, resulting in an H-Score of 62.5, significantly outperforming OrthogonalDet's H-Score of 34.8. This substantial improvement validated that EUS effectively detected unknown objects and distinguished them well from known objects even in the remote sensing domain. Throughout incremental tasks, the known mAP remained consistently high across tasks. By Task 4, DEUS achieved 68.3 compared to OrthogonalDet's 64.2 while consistently maintaining high unknown recall performance. This indicates that EKD successfully mitigated catastrophic forgetting in the remote sensing domain. These results demonstrated that DEUS provided a generalizable solution for OWOD that extended beyond natural image domains, making it applicable to specialized applications such as remote sensing, medical imaging, and autonomous navigation systems.

# D  QUALITATIVE RESULTS

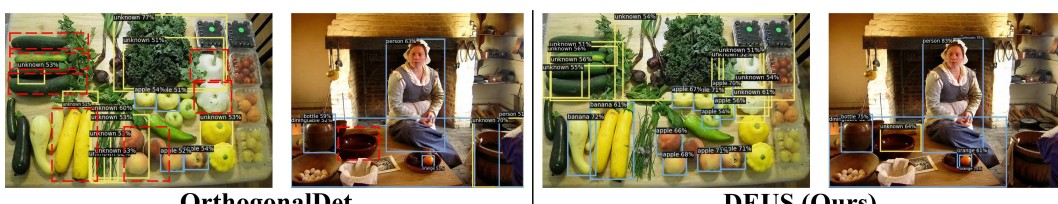

**OrthogonalDet**      **DEUS (Ours)**

S.Figure 1: Qualitative results after training on Task 3. Blue and yellow boxes indicate the predictions for known and unknown objects. Red boxes indicate missed objects, which may belong to either known or unknown categories.

We conducted qualitative experiments on images containing various objects and compared the results with OrthogonalDet. The visualizations were generated using the model continually trained on Task 3 in M-OWODB, utilizing the MS-COCO dataset. Blue and yellow boxes represent known and unknown objects, respectively, while red-dotted boxes indicate missed objects that belong to either known or unknown categories. As shown in S.Figure 1, OrthogonalDet failed to detect certain known objects *e.g.*, banana and apple in the left image as well as unknown objects, whereas the proposed DEUS effectively detected both known and unknown objects.

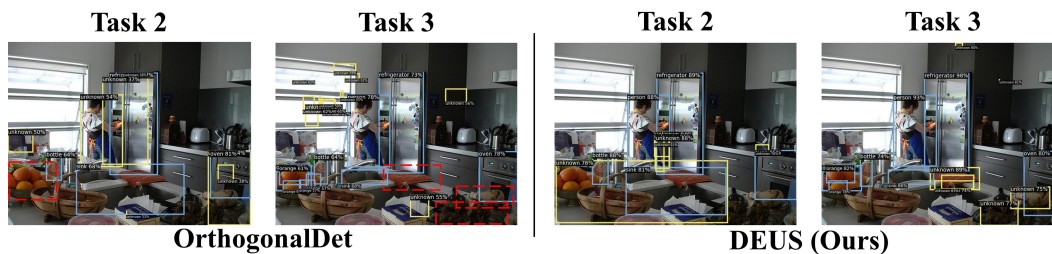

**Task 2** **Task 3** **Task 2** **Task 3**

**OrthogonalDet** **DEUS (Ours)**

S.Figure 2: Qualitative Results of Continuous Training on Tasks in M-OWODB. Blue and yellow boxes represent predictions for known and unknown objects, respectively. Red boxes indicate missed objects, which may belong to either known or unknown categories. Task 2 includes appliance classes such as refrigerator, sink, and oven, while Task 3 includes fruit classes like orange.

Additionally, we conducted qualitative experiments to evaluate the ability to continually detect objects in the same input image. We visualized the results after training on Task 2 and Task 3. Task 2 includes appliance classes such as refrigerator, sink, and oven, while Task 3 includes fruit classes such as orange. As shown in S.Figure 2, OrthogonalDet failed to detect the orange as an unknown object, whereas DEUS correctly labeled it as unknown. After training on Task 3, both OrthogonalDet and DEUS were able to detect the newly learned orange as a known class. However, while OrthogonalDet failed to detect unknown objects and mistakenly labeled backgrounds as unknown, DEUS effectively detected unlabeled unknown objects. These qualitative results validate that the proposed DEUS enables the detector to correctly identify unknown objects even in images containing a variety of objects.