# OpenReview forum: "Detecting Unknown Objects via Energy-based Separation for Open World Object Detection"
_ICLR.cc/2026/Conference — ICLR 2026 Conference Withdrawn Submission_

### Official Review · Reviewer_141J · 2025-10-27

**Soundness:** 3
**Presentation:** 2
**Contribution:** 3
**Rating:** 6
**Confidence:** 4

**Summary:**

This paper presents DEUS, an energy-based framework for open-world object detection (OWOD). It introduces two key components: Energy-based orthogonal subspace separation (EUS) to geometrically distinguish known and unknown regions in the feature space, and Energy-based Known Distinction (EKD) loss to mitigate catastrophic forgetting during incremental learning. The approach achieves strong performance across multiple OWOD benchmarks, offering both theoretical grounding and solid empirical results.

**Strengths:**

1. Sound theoretical motivation: The combination of energy modeling and geometric subspace constraints provides a clear and interpretable foundation for separating known and unknown classes.
2. Practical effectiveness: EKD loss successfully stabilizes continual learning and alleviates old-class degradation.
3. Comprehensive experimentation: Evaluations are broad and convincing, showing consistent improvements in both unknown recall and overall H-score.
4. Stable and reproducible: The framework remains within a well-understood paradigm, enhancing reliability and implementation clarity.

**Weaknesses:**

1. Incremental innovation: While the integration of energy modeling and ETF subspaces is elegant, it largely extends existing formulations rather than defining a new paradigm.
2. Method complexity: The combination of multiple subspaces and loss terms increases the model’s training and inference cost; efficiency analysis is missing.
3. Limited semantic reasoning: Unknowns are modeled only in feature/energy space, without deeper semantic interpretation or taxonomy-level analysis.
4. EKD generalization: The method’s applicability beyond OWOD (e.g., classification or segmentation) is not verified.

**Questions:**

1. Efficiency and Scalability:
Could you provide a quantitative comparison of training/inference time and GPU memory with OW-DETR or PROB? This would clarify whether the added subspace and loss terms impose significant overhead.
2. Ablation on Subspace Orthogonality:
How sensitive is EUS to the strength of orthogonality constraints? Would partial relaxation (non-orthogonal but low-correlation spaces) harm performance?
3. Semantic Modeling of Unknowns:
Have you considered integrating higher-level cues (e.g., attribute clusters, language priors) into the unknown modeling process to enhance semantic interpretability?
4. Generalization of EKD:
Do you believe EKD could be easily transferred to other continual learning tasks (classification, segmentation)? Any early observations or limitations to share?

---

### Official Review · Reviewer_BJC4 · 2025-10-30

**Soundness:** 3
**Presentation:** 2
**Contribution:** 2
**Rating:** 2
**Confidence:** 3

**Summary:**

The authors have proposed a framework DEUS for Open World Object Detection that consists of ETF-Subspace Unknown Separation (EUS) and an Energy-based Known Distinction (EKD) loss.
Experimental results demonstrate this method’s ability to distinguish known from unknown objects. However, the two modules lack innovation, as they are merely a simplistic combination of existing techniques.

**Strengths:**

1. The proposed model first utilizes EUS to separate the known and unknown feature spaces, and then employs the EKD loss to overcome the key shortcoming of prior energy-based methods: their exclusive focus on energy within the known space.

2. The comparison of the model with the latest methods demonstrates its outstanding performance.

3. The model structure is described in the article with sufficient detail.

**Weaknesses:**

1. EUS and the EKD loss have been extensively applied across various fields. Although the authors introduce a new network architecture to first decouple the known and unknown spaces and then calculate energy scores, the conceptual advance over existing approaches is limited and not clearly demonstrated.

2. The research motivation for designing EUS to achieve feature space decoupling is inadequately described, making it unclear how EUS advances beyond other decoupling methods. The use of fixed and non-learnable bases appears to be directly adopted rather than original.

3. While the paper is technically sound, it fails to demonstrate significant innovation. The formulas and methodologies presented, though competently described, rely on well-established techniques rather than introducing novel concepts.

4. The ablation studies are insufficient.

**Questions:**

1. Where does the novelty of EUS and the EKD loss lie, specifically?​​ Given their prior applications in other fields, what constitutes the genuine conceptual advancement in this work? What components are genuinely redesigned compared to existing methods?​​ For instance, is the fundamental computational paradigm new, or is it primarily an application of established techniques to a different problem?

2. What is the demonstrated advantage of EUS over other feature separation or decoupling methods?​​ Are there comparative experimental results that quantitatively validate its superiority in the context of open-world detection?

3. Which formulas in the paper have been genuinely redesigned? Most of the formulations fail to demonstrate technical innovation. It is recommended that the authors remove certain well-established components, such as the focal loss for classification.

4. Since the method employs multiple loss functions, it is recommended that the authors supplement additional experiments to ablate the contributions of key components, such as the energy-based margin loss or the two parts of the EKD loss.

---

### Official Review · Reviewer_6FRr · 2025-11-01

**Soundness:** 3
**Presentation:** 2
**Contribution:** 2
**Rating:** 4
**Confidence:** 3

**Summary:**

This work addresses the challenging task of Open World Object Detection (OWOD), where the model must incrementally learn to classify known objects without forgetting, while simultaneously identifying previously unseen objects without supervision. The proposed method, termed Detecting Unknowns via Energy-based Separation (DEUS), introduces two key components: ETF-Subspace Unknown Separation (EUS) and Energy-based Known Distinction (EKD) loss. EUS uses ETF( Equiangular Tight Frame (ETF) )-based geometric properties to construct orthogonal subspaces that facilitate the separation of unknown objects, while the EKD loss encourages clear distinction between classifiers from previous and current training stages. Experimental results demonstrate that DEUS improves unknown object recall while maintaining competitive performance on known classes across both M-OWODB and S-OWODB benchmarks.

**Strengths:**

The paper addresses a challenging problem in OWOD.

Using a Simplex ETF to geometrically separate known and unknown feature spaces is an interesting approach.

The reported improvements in unknown recall, relative to baseline models, are particularly promising.

**Weaknesses:**

It is not clear why fixed simplex-ETF subspaces are expected to yield effective separation between known and unknown categories.

Additionally, the method appears complex, relying on multiple components; the overall pipeline feels heavy, and the ablation analysis does not clearly isolate the contribution of each component.

The ETF concept is introduced only in the Method section; it would be beneficial to briefly explain the term in the Introduction for better clarity.

**Questions:**

The method is closely coupled with OrthogonalDet as the underlying detector. Although this is a sensible choice for OWOD, such strong dependence on its architectural design raises concerns regarding the method’s applicability to alternative detection backbones.

---

### Official Review · Reviewer_Y92H · 2025-11-06

**Soundness:** 3
**Presentation:** 3
**Contribution:** 3
**Rating:** 6
**Confidence:** 4

**Summary:**

The paper identifies two problems in the existing Open-World Object Detection (OWOD) works: (i) Unknown objects getting confused with background regions, and (ii) Cross-influence between old and new known classes during incremental training. Figure 1 demonstrates these problems. To address them, the paper proposes DEUS, which has two components EUS and EKD, that tackle the issues (i) and (ii) respectively. EUS constructs two distinct Simplex-ETF subspaces for known and unknown classes, guiding training such that known and unknown samples exhibit high energy in their respective subspaces while background proposals are pushed toward the boundary region between them. EKD divides the known classifier into old and new sub-classifiers and, analogous to EUS, encourages higher energy alignment between samples and their corresponding classifier. Combined, EUS and EKD improve unknown class discovery and reduce forgetting in known classes over the incremental tasks.

**Strengths:**

* The paper identifies two important issues in the existing OWOD works and proposes complementary modules to address them.
* Energy-based contrastive formulation is creatively utilized in both EUS and EKD to separate knowns from unknowns and new knowns from old knowns respectively. EUS contrasts known vs. unknown subspaces to improve unknown discovery while EKD contrasts old vs. new sub-classifiers to mitigate forgetting.
* The paper achieves strong H-scores across tasks and benchmarks that shows its ability in improving unknown discovery while mitigating known-class forgetting.
* The visualizations are clear, well-designed, and effectively convey the key ideas

**Weaknesses:**

* Results emphasize U‑Recall and H‑Score for unknown classes. Precision-oriented unknown metrics (e.g., Unknown Detection Precision (UDP)) are absent; high recall can coincide with many false unknowns. Including these would better characterize trade‑offs.
* Supplementary lists a 3×4090 setup and proposal counts, but the paper lacks wall‑clock, FLOPs, or memory overhead analyses for the EUS/EKD modules. EUS introduces extra projections onto 128 ETF vectors; even if cheap, runtime/throughput numbers would be useful.
Although the supplementary describes a more selective pseudo-labeling scheme (threshold at final unknown logit > 0, proportional count, size filtering), there is limited ablation on the threshold and size filters, which could materially impact unknown quality and hence the EUS‑driven loop.

Minor:
* The paper needs a proof-reading for grammatical mistakes, eg: line 425-426 - resulting in significantly improved H-Scores that demonstrating … - remove ‘that’ or replace demonstrating with demonstrates.
* The paper needs to include the following sections as per ICLR guidelines: Reproducibility Statement, Limitations, and Declaration of LLM usage.

**Questions:**

* While the proposed method achieves high U-recall, this might also coincide with many false unknowns. Although not highly adapted in recent works, a 2022 paper titled ”Revisiting Open World Object Detection” introduces the metric Unknown Detection Precision (UDP). Studying this metric would strengthen the claims of the proposed work further.
* In the pseudo‑labeling process, how does varying the “> 0” unknown‑logit threshold or the bbox size filter affect U‑Rec and known mAP? A small ablation in the supplement would be valuable.
* What is the memory‑buffer size and sampling policy for replay? Are all methods evaluated under the same memory budget and sampling? Including this would improve fairness.
* What is the inference throughput change vs. OrthogonalDet (proposals/sec, ms/image)?

---

### Note · Authors · 2025-11-14

I have read and agree with the venue's withdrawal policy on behalf of myself and my co-authors.